# Slow-wave sleep drives sleep-dependent renormalization of synaptic AMPA receptor levels in the hypothalamus

Jianfeng Liu[1], Niels Niethard[1], Yu Lun[1], Stoyan Dimitrov[1], Ingrid Ehrlich[2], Jan Born [1,3,4,5,6]*, Manfred Hallschmid [1,4,5,6]*

1 Institute of Medical Psychology and Behavioural Neurobiology, University of Tübingen, Tübingen, Germany, 2 Department of Neurobiology, Institute of Biomaterials and Biomolecular Systems, University of Stuttgart, Stuttgart, Germany, 3 Center for Integrative Neuroscience, University of Tübingen, Tübingen, Germany, 4 German Center for Diabetes Research (DZD), Tübingen, Germany, 5 Institute for Diabetes Research and Metabolic Diseases of the Helmholtz Center Munich at the University Tübingen (IDM), Tübingen, Germany, 6 German Center for Mental Health (DZPG), Tübingen, Germany

☯ These authors contributed equally to this work.
* jan.born@uni-tuebingen.de (JB); manfred.hallschmid@uni-tuebingen.de (MH)

**Data Availability Statement:** All relevant data are within the paper and its Supporting Information

## Abstract

According to the synaptic homeostasis hypothesis (SHY), sleep serves to renormalize synaptic connections that have been potentiated during the prior wake phase due to ongoing encoding of information. SHY focuses on glutamatergic synaptic strength and has been supported by numerous studies examining synaptic structure and function in neocortical and hippocampal networks. However, it is unknown whether synaptic down-regulation during sleep occurs in the hypothalamus, i.e., a pivotal center of homeostatic regulation of bodily functions including sleep itself. We show that sleep, in parallel with the synaptic down-regulation in neocortical networks, down-regulates the levels of α-amino-3-hydroxy-5-methyl-4-isoxazolepropionic acid receptors (AMPARs) in the hypothalamus of rats. Most robust decreases after sleep were observed at both sites for AMPARs containing the GluA1 subunit. Comparing the effects of selective rapid eye movement (REM) sleep and total sleep deprivation, we moreover provide experimental evidence that slow-wave sleep (SWS) is the driving force of the down-regulation of AMPARs in hypothalamus and neocortex, with no additional contributions of REM sleep or the circadian rhythm. SWS-dependent synaptic down-regulation was not linked to EEG slow-wave activity. However, spindle density during SWS predicted relatively increased GluA1 subunit levels in hypothalamic synapses, which is consistent with the role of spindles in the consolidation of memory. Our findings identify SWS as the main driver of the renormalization of synaptic strength during sleep and suggest that SWS-dependent synaptic renormalization is also implicated in homeostatic control processes in the hypothalamus.

files. All custom MATLAB codes are available at https://zenodo.org/records/12771672.

**Funding:** This research was supported by grants from the Deutsche Forschungsgemeinschaft to J. B. and I.E. (https://dfg.de/en; FOR 5434), from the European Research Council to J.B. (https://erc.europa.eu/homepage; ERC AdG 883098 SleepBalance), from the German Federal Ministry of Education and Research (BMBF; https://bmbf.de/bmbf/en) to the German Center for Diabetes Research (DZD e.V.; https://www.dzd-ev.de/en; 01GI0925), and from the Hertie Foundation, Network for Excellence in Clinical Neuroscience, to N.N. (https://www.ghst.de/en/studying-the-brain/creating-structures/hertie-network-of-excellence-in-clinical-neuroscience). J.L. and Y.L gratefully acknowledge funding from the China Scholarship Council (CSC; https://chinesescholarshipcouncil.com; # 201506180020 and # 201808080072). The funders had no role in study design, data collection and analysis, decision to publish, or preparation of the manuscript.

**Competing interests:** The authors have declared that no competing interests exist.

**Abbreviations:** AMPAR, α-amino-3-hydroxy-5-methyl-4-isoxazolepropionic acid receptor; CDK5, cyclin-dependent kinase 5; EEG, electroencephalographic; EMG, electromyographic; LTD, long-term depression; LTP, long-term potentiation; PBS, phosphate-buffered saline; PKA, protein kinase A; REM, rapid eye movement; REM-D, REM sleep deprivation; SD, standard deviation; SHY, synaptic homeostasis hypothesis; SO, slow oscillation; SWS, slow-wave sleep; TSD, total sleep deprivation.

## Introduction

Experience during the wake phase is encoded into the brain's neuronal networks by strengthening the synaptic connections in specific neuron ensembles. The encoding of information during wakefulness thus leads to a widespread strengthening of synaptic networks, which, in the absence of counterregulatory processes, would ultimately yield a state of saturation. The synaptic homeostasis theory (SHY) proposes that sleep following the wake phase is the essential process that broadly renormalizes synaptic strength [1,2]. Specifically, SHY assumes that wake encoding of information manifests itself mainly in the potentiation of excitatory synapses, which is known to result in increased numbers of α-amino-3-hydroxy-5-methyl-4-isoxazolepropionic acid (AMPA) type glutamate receptors, while they are down-regulated during subsequent sleep. This down-selection of potentiated synapses is assumed to be driven by the <1 Hz slow oscillation (SO) that hallmarks the stage of slow-wave sleep (SWS). SOs might generally weaken synaptic strength by decreasing synchronized firing, except in those neuron ensembles that were essentially involved in prior learning and, during subsequent sleep, are reactivated in the excitable up-states of the SO [3–5]. However, there is also evidence that the renormalization of potentiated synapses primarily occurs during theta activity, a key phenomenon of rapid eye movement (REM) sleep [6,7]. SHY has been supported by numerous functional and structural studies of synaptic networks in the neocortex and hippocampus, which represent the main building blocks of the episodic memory system [8–11].

Surprisingly, previous work has entirely ignored the question whether sleep also renormalizes synaptic networks in the hypothalamus, a brain region pivotal for the homeostatic regulation of multiple organismic processes including metabolic and reproductive functions, as well as circadian and sleep/wake rhythms. In principle, such bodily homeostatic control processes could be established in the absence of encoding-related synaptic upscaling; consequently, they would not be in need of any sleep-dependent processes of synaptic renormalization. Alternatively, whole-body homeostatic control in hypothalamic networks may be indispensably bound to the continuous encoding and integration of environmental information during the wake phase; it would therefore imply the upscaling of synaptic networks and, as a consequence, the need for synaptic renormalization during sleep. In line with the latter assumption, there is growing evidence suggesting that hypothalamic circuits balancing such bodily functions exhibit plasticity involving glutamatergic neurotransmission [12–15]. Thus, challenging homeostatic regulation by high-fat feeding or osmotic salt loading invokes distinct changes in glutamatergic neurotransmission and associated synaptic expression of AMPARs in local circuits and wider networks in the hypothalamus [16,17]. Moreover, recent studies point to a critical role of the hypothalamus in learning and the formation of persisting memory representations for social and nonsocial (spatial, object) experiences [18–21].

We investigated the role of sleep, and of distinct sleep stages, in the down-regulation of excitatory glutamatergic synapses in the rat hypothalamus. To corroborate and relate our results to previous findings [8], we also assessed respective changes in the neocortex. We focused on postsynaptic AMPARs, specifically on the GluA1 and GluA2 subunits and the phosphorylation of GluA1 in synaptoneurosomes, as key substrates of synaptic renormalization and plasticity at excitatory synapses [22–24]. These 2 subunits are also the most prominent AMPAR subunits expressed in rat hypothalamus and neocortex [15]. We conducted 2 independent experiments to disentangle sleep-specific from potential circadian effects. In experiment 1, we compared a spontaneous wake group, in which tissue for synaptoneurosome analyses was obtained at 24:00 h of the active phase, with a sleep group, in which tissue was obtained at 12:00 h of the inactive phase, resulting in a between-group circadian shift of 12

hours. In experiment 2, we compared 3 groups, i.e., sleep, total sleep deprivation (TSD), and REM sleep deprivation (REM-D), which were all killed at precisely the same time as the sleep group of experiment 1 (12:00 h of the inactive phase), effectively eliminating any circadian influence. We found that independent of the circadian rhythm, sleep compared to wakefulness leads to a distinct decrease in the synaptic AMPAR subunit GluA1 in the hypothalamus and that the reductions in hypothalamic GluA1 subunits due to sleep parallel those found in neocortex. Sleep-associated attenuations of GluA1-containing AMPARs phosphorylated at Ser845 or Ser831 appeared to be generally enhanced by circadian rhythmicity. Decreases in AMPAR subunit expression were comparable after undisturbed sleep and selective deprivation of REM sleep, indicating that SWS is the main driver of synaptic down-regulation in hypothalamic networks.

## Results

### Diminished AMPAR expression in hypothalamus and neocortex after sleep compared with wakefulness

We measured the expression of AMPAR subunits in synaptoneurosomes obtained in the entire hypothalamus and in the left cortical hemisphere in 2 groups of rats after 6-hour periods, which took place either during the animals' daytime rest period or nighttime active period and, accordingly, were filled with spontaneous sleep (Sleep; $n$ = 16) or spontaneous wakefulness (Wake; $n$ = 16; see **Fig 1A** for the design of experiment 1). Synaptoneurosomes are enriched in synaptic proteins and, thus, optimal for detecting activity-dependent changes in glutamate receptor levels (see also Methods/Preparation of synaptoneurosomes and **S1 Fig**). Time spent asleep during the 6-hour interval was (mean ± SEM) 202.68 ± 15.33 min in the Sleep group and 94.93 ± 5.96 min in the Wake group (t(30) = −6.55, $p < 0.001$; **Fig 1B**).

In the hypothalamus, expression of AMPAR subunits in synaptoneurosomes was generally lower after sleep than after wakefulness (**Fig 1C**): relative to the Wake group, animals of the Sleep group showed a decrease to 57.5 ± 4.3% of GluA1 levels in synaptoneurosomes (t(30) = 5.489, $p < 0.001$, mean Wake value set to 100%). The change in GluA2 levels to 88.5 ± 15.0% was not significant ($p = 0.577$). Compared with the Wake rats, the Sleep rats also showed decreased levels of GluA1 phosphorylated at Ser845 (61.0 ± 5.9%, t(30) = 3.517, $p < 0.01$; **Fig 1D**), whereas the respective pattern in GluA1 phosphorylated at Ser831 did not reach significance (91.3 ± 8.3%, $p = 0.495$).

In the neocortex, we detected differences between the Sleep and Wake groups in the levels of GluA1- and GluA2-containing AMPARs in synaptoneurosomes and in GluA1 phosphorylated at Ser845 and Ser831. Compared with the Wake group, the Sleep group showed a decrease in the cortical levels of AMPAR-containing GluA1 (63.1 ± 3.9%, t(30) = 2.873, $p < 0.05$) and GluA2 (83.7 ± 3.7%, t(30) = 2.674, $p < 0.05$; **Fig 1E**), as well as distinct decreases in the levels of GluA1 phosphorylated at Ser845 (55.8 ± 4.0%, t(30) = 2.674, $p < 0.001$) and Ser831 (71.9 ± 7.2%, t(30) = 2.960, $p < 0.01$; **Fig 1F**). Direct statistical comparisons between neocortex and hypothalamus of the relative decreases in protein levels of AMPAR subunits after sleep versus wakefulness revealed the sleep-related decreases to be comparable between both sites for all 4 protein measures (F(1,18) < 0.445, $p > 0.51$ for respective Sleep/ Wake × Neocortex/Hypothalamus interactions). Control analyses of AMPAR subunit levels in supernatants did not indicate any detectable differences between the Sleep and Wake groups in hypothalamic or cortical samples (**S2 Fig**), confirming that the observed sleep-associated decreases in synaptoneurosomal GluA1 and GluA2 subunits are specific to the synapses. Importantly, to exclude any potential confounding effect of normalization to β-actin bands,

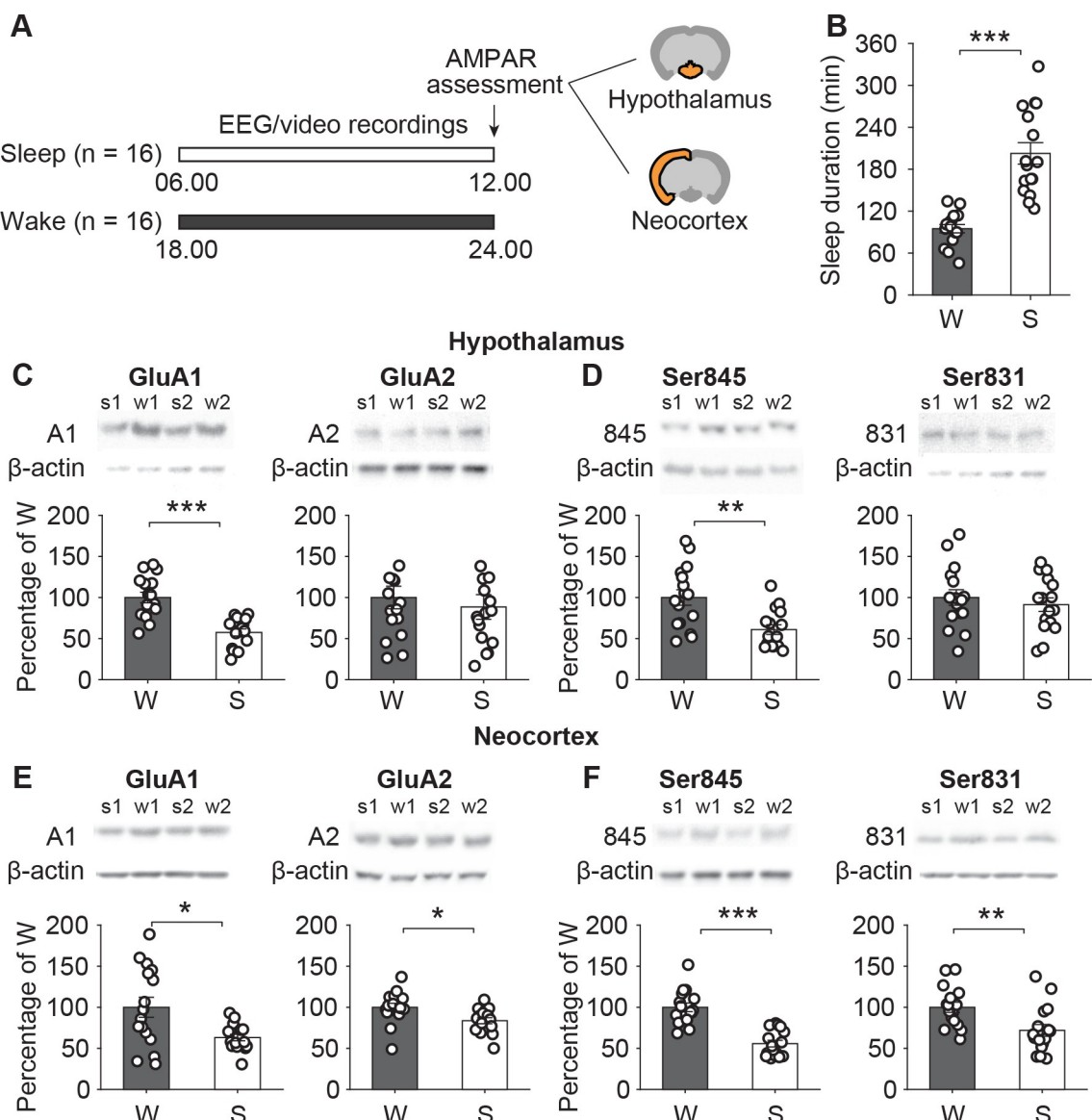

**Fig 1. AMPAR levels in hypothalamus and neocortex after sleep and wakefulness.** (**A**) Experimental design: AMPAR subunit levels were measured in synaptoneurosomes sampled from the entire hypothalamus and from the left cortical hemisphere of rats after experimental 6-hour periods taking place either during the animals' daytime rest period (starting at 6:00 h, lights on) or nighttime active period (starting at 18:00 h, lights off) and, accordingly, filled with spontaneous sleep (Sleep group, S; $n = 16$; white bars) or with wakefulness (Wake group, W; $n = 16$; black bars). During the 6-hour interval, animals were food-deprived while water was available ad libitum, and sleep was assessed by visually scoring behavior (in 6 rats) and by EEG and EMG recordings (in 10 rats). (**B**) Mean ($\pm$ SEM) time (in min) spent asleep during the 6-hour interval before AMPAR assessment (dot plots overlaid). (**C**) Levels of GluA1- (left) and GluA2-containing AMPARs (right) and (**D**) of GluA1 phosphorylated at Ser845 (left) and at Ser831 (right) in hypothalamus and (**E/F**) neocortex. For AMPAR subunit levels, mean $\pm$ SEM normalized values are shown with means of the Wake group set to 100%. On top of the panels, 2 example immunoblots are shown for each group (s1, s2, w1, w2; GluA1, GluA2, phospho-Ser845, and phospho-Ser831 bands were normalized with reference to the corresponding β-actin band in the same sample, the latter serving as loading control). * $p < 0.05$, ** $p < 0.01$, *** $p < 0.001$, unpaired $t$ tests; the underlying data sets are available in an online supporting file (S1 Data).

we also directly compared β-actin levels collected from synaptoneurosomes and found that they did not differ between groups (hypothalamus: t(78) = 0.531, $p > 0.597$; neocortex: t(78) = −1.318, $p > 0.191$).

## AMPAR expression is higher after total sleep deprivation but unchanged after selective REM sleep deprivation compared to sleep

In order to investigate the possible causal contribution of REM sleep to the down-regulation of AMPARs, we performed a second experiment in 3 groups of rats (**Fig 2A**), which were exposed to total sleep deprivation for the entire experimental 6-hour period before AMPAR assessment (TSD), or to REM sleep deprivation by gentle handling during the 6-hour period (REM-D), or whose sleep was not disturbed (Sleep, S; $n$ = 8 rats in each group). For all groups, the 6-hour period before AMPAR assessment started at 6:00 h (lights on), i.e., with the beginning of the rest period. Total sleep deprivation reduced sleep time to a minimum of 0.47% of that of the Sleep group (**Fig 2B**). REM sleep deprivation suppressed REM sleep to 0.78% of that during undisturbed sleep in the Sleep group, while total sleep time and SWS duration in the REM-D group did not significantly differ from the Sleep group (**Fig 2B**).

Hypothalamic levels of the AMPAR GluA1 subunit were increased after total sleep deprivation to 142.7 ± 14.1% of those in the Sleep group (set to 100%; t(16) = −2.685, $p$ = 0.018; **Fig 2C**). After REM sleep deprivation, levels of GluA1-containing AMPARs in hypothalamic synaptoneurosomes were closely comparable to those found after undisturbed sleep in the Sleep group (t(16) = 0.015, $p$ = 0.989) and, consequently, also significantly lower than after total sleep deprivation (t(16) = 2.575, $p$ = 0.022; F(2,21) = 5.518, $p$ = 0.012 for main effect of Sleep versus TSD versus REM-D). We did not find significant differences between any 2 of the groups in GluA2 subunit levels (F(2,21 = 0.082, $p$ = 0.921; **Fig 2C**). There were also no differences between the 3 groups in GluA1 subunits phosphorylated at Ser845 (F(2, 21) = 0.07, $p$ = 0.933) or at Ser831 (F(2,21) = 0.707, $p$ = 0.504; **Fig 2D**).

In the neocortex, GluA1 levels were likewise highest after total sleep deprivation (136.5 ± 5.2%, with levels of the Sleep group set to 100%). Thus, they were not only significantly higher than levels in the Sleep group (t(16) = 3.768, $p$ = 0.002) but also higher than levels in the REM sleep-deprived rats (145.3 ± 5.5%, t(16) = 3.248, $p$ = 0.009; F(2,21) = 6.665, $p$ = 0.006 for main effect; **Fig 2E**). GluA1 levels in REM-D and Sleep rats were almost identical (t(16) = 0.414, $p$ = 0.685). In parallel, levels of GluA1 subunits phosphorylated at Ser845 were enhanced after TSD in comparison both with Sleep (139.92 ± 8.3%, t(16) = −2.435, $p$ = 0.029) and with REM-D (145.43 ± 8.6%, t(16) = 2.601, $p$ = 0.021; F(2,21) = 3.644, $p$ = 0.044, for main effect; **Fig 2F**). There were no differences in neocortical synaptosomes in the levels of GluA1 subunits phosphorylated at Ser831 (F(2,21) = 1.547, $p$ = 0.236) or GluA2-subunits of AMPARs (F(2,21 = 1.593, $p$ = 0.227; **Fig 2E and 2F**). Comparisons of changes in AMPAR subunits found in hypothalamic versus neocortical synaptosomes did not yield any significant sleep-dependent differences between the 2 sites ($p$ > 0.289 for respective TSD/REM-D/ Sleep × Neocortex/Hypothalamus interactions).

Control analyses of supernatants and β-actin levels did not reveal any differences between the 3 groups, corroborating that the observed changes in AMPAR levels were specific to the synaptoneurosomes (**S3 Fig**) and, furthermore, not related to changes in β-actin levels (hypothalamus: F(2,93) = 0.099, $p$ > 0.905); neocortex: F(2,93) = 1.100, $p$ > 0.336).

In experiment 1, the 6-hour experimental sleep and wake periods took place during the animals' rest and activity phases, respectively, and thus confounded effects of sleep and circadian rhythm, whereas in experiment 2, the experimental 6-hour periods before AMPAR assessment took place at the same circadian phase in all experimental groups, i.e., between 6:00 and 12:00 h. Accordingly, additional analyses comparing the effects of undisturbed sleep relative to wakefulness (i.e., with the values of the Wake and TSD groups in experiments 1 and, respectively, 2, set to 100%), which covered experiments 1 versus 2 and both sites of interest (hypothalamus versus neocortex), allowed the differentiation of potential circadian influences from effects of sleep.

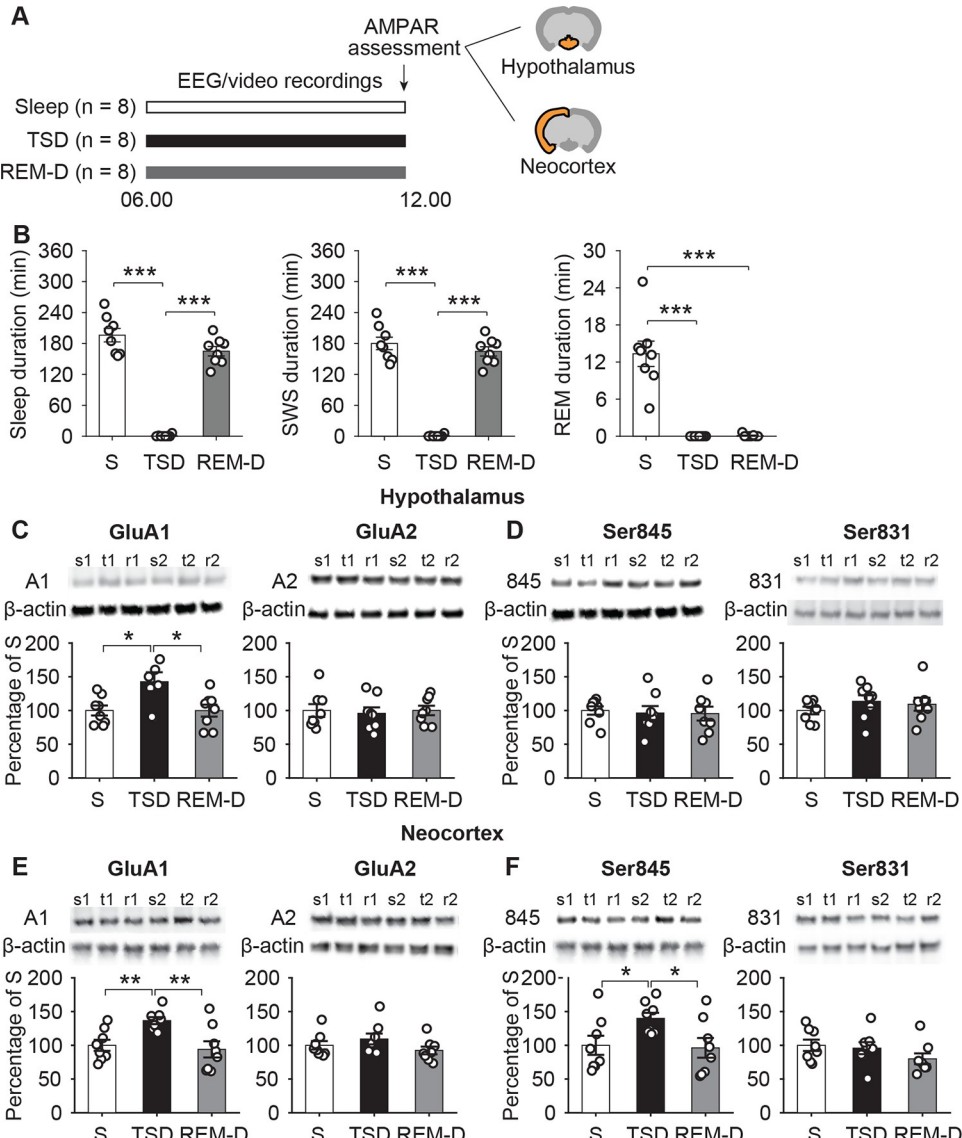

**Fig 2. Changes in AMPAR subunit levels after undisturbed sleep (S), total sleep deprivation (TSD), and REM sleep deprivation (REM-D).** (**A**) Study design: 3 groups of rats were compared, i.e., a Sleep group with undisturbed sleep during the 6-hour period before AMPAR assessment (S; $n$ = 8 rats; white bars), a total sleep deprivation group, which was kept awake during the 6-hour period (TSD; $n$ = 8 rats; black bars), and a REM sleep deprivation group that was selectively deprived of REM sleep during the 6-hour period (REM-D; $n$ = 8 rats; grey bars). The experimental 6-hour period started always at 6:00 h and, as in experiment 1, included food but not water deprivation. Sleep deprivation was achieved by gentle handling. (**B**) Mean ± SEM time (in min) spent in sleep (left), SWS (middle), and REM sleep (right) by the 3 groups (dot plots overlaid). (**C**) Levels of GluA1 (left) and GluA2 AMPAR subunits (right) and (**D**) of GluA1 subunits phosphorylated at Ser845 (left) and at Ser831 (right) in hypothalamus and (**E/F**) neocortex. For AMPAR subunit levels, mean ± SEM normalized values are shown with means of the Sleep group set to 100% (dot plots overlaid). On top of panels, 2 example immunoblots are shown for each group (s1, s2, t1, t2, r1, r2; GluA1, GluA2, phospho-Ser845, and phospho-Ser831 bands were normalized with reference to the corresponding β-actin band in the same sample, the latter serving as loading control). * $p < 0.05$, ** $p < 0.01$, *** $p < 0.001$, unpaired $t$ tests; the underlying data sets are available in an online supporting file (S1 Data).

These ANOVA comparisons did not yield robust differences between experiments in the sleep effect on GluA1 and GluA2 levels (all $p > 0.065$) but revealed a main effect of Experiment (1 versus 2) for the levels of GluA1 subunits phosphorylated at Ser845 and Ser831 (F(1,16) =

14.804, $p < 0.01$ and $F(1,16) = 9.157$, $p < 0.01$, respectively; $p > 0.106$ for respective Experiment × Hypothalamus/Neocortex interactions). Post hoc analyses confirmed more pronounced decreases (compared to Wake and, respectively, TSD) in the Sleep groups of experiment 1 than experiment 2 for hypothalamic levels of GluA1 phosphorylated at Ser845 ($60.930 ± 8.705\%$ versus $103.846 ± 6.442\%$, $t(16) = −3.963$, $p < 0.01$) and for neocortical levels of GluA1 phosphorylated at Ser831 ($67.42 ± 8.145\%$ versus $104.432 ± 8.807\%$, $t(16) = −3.085$, $p < 0.01$). This pattern indicates a predominant role of sleep rather than circadian factors for the regulation of GluA1 and GluA2 levels, whereas circadian rhythmicity might add to the effect of sleep on GluA1 subunit phosphorylation.

## Spindle density predicts levels of GluA1-containing AMPARs in hypothalamic synaptoneurosomes

We next investigated which neurophysiological features of sleep may shape the configuration of hypothalamic AMPARs. For this purpose, we analyzed the association of oscillatory sleep features with AMPAR expression at the end of the experimental sleep period in the rats that slept undisturbed while continuous electroencephalographic (EEG; from scull electrodes) and electromyographic signals (EMG) were recorded. This was the case in 10 rats of the Sleep group of experiment 1 and in all rats of the Sleep group in experiment 2.

Analyses of experiment 1 were run in an exploratory fashion and aimed to identify EEG oscillatory features predicting AMPAR levels, and we used analyses of experiment 2 to rebut or confirm these results. We focused on amplitude and duration of EEG oscillations that are known to contribute to synaptic plasticity processes during sleep, i.e., on 0.1 to 4 Hz slow-wave activity and 10 to 16 Hz spindles as hallmarks of SWS, and on 5 to 10 Hz theta activity as a key characteristic of REM sleep [7,25,26]. After an initial analysis of correlations between AMPAR subunit levels and sleep parameters for the entire experimental 6-hour interval preceding AMPAR assessment, we ran separate analyses for the first and second 3-hour intervals, taking into account that total sleep time and time spent in SWS and REM sleep increased from the first to the second 3-hour interval (**Figs 3A-3C** and **S4**). Only a few of the sleep parameters assessed in experiment 1 showed consistent associations with AMPAR subunit expression, and only with GluA1 subunit levels (see **S1 Table** for a summary of results). Thus, levels of GluA1-containing AMPARs in hypothalamic synaptoneurosomes were positively correlated with spindle density, i.e., sleep with high spindle density was associated with enhanced GluA1 subunit levels. This correlation approached significance for the 6-hour interval ($r = 0.6135$, $p = 0.0593$) and was significant for the late 3-hour interval ($r = 0.6791$, $p = 0.0308$; **Fig 3D**). Notably, whereas GluA1-containing AMPAR levels were positively associated with spindle density, they were negatively correlated with SWS duration, i.e., were lower when rats spent more time in SWS during the last 3-hour period ($r = −0.729$, $p = 0.0168$). None of the other SWS-related parameters showed any consistent relationship with AMPAR levels (all $p > 0.095$). However, levels of GluA1-containing AMPARs in hypothalamic synaptosomes were also negatively correlated with REM sleep theta energy during the 6-hour interval ($r = −0.846$, $p = 0.0020$). This negative correlation was even more pronounced in the separate analysis of the second 3-hour interval ($r = −0.8740$, $p = 0.0009$), where it also reached significance for REM sleep duration ($r = −0.6471$, $p = 0.0431$; **S1 Table**). For AMPAR levels in cortical synaptoneurosomes, we did not find any consistent associations with SWS- or REM sleep-related parameters during the first and last 3 hours of sleep before their assessment (**S1 Table**).

In order to confirm or rebut spindle density, time spent in SWS, and REM sleep theta energy as predictors of subunit levels in hypothalamic synaptoneurosomes, we analyzed data from the Sleep group of experiment 2, where animals likewise experienced uninterrupted sleep

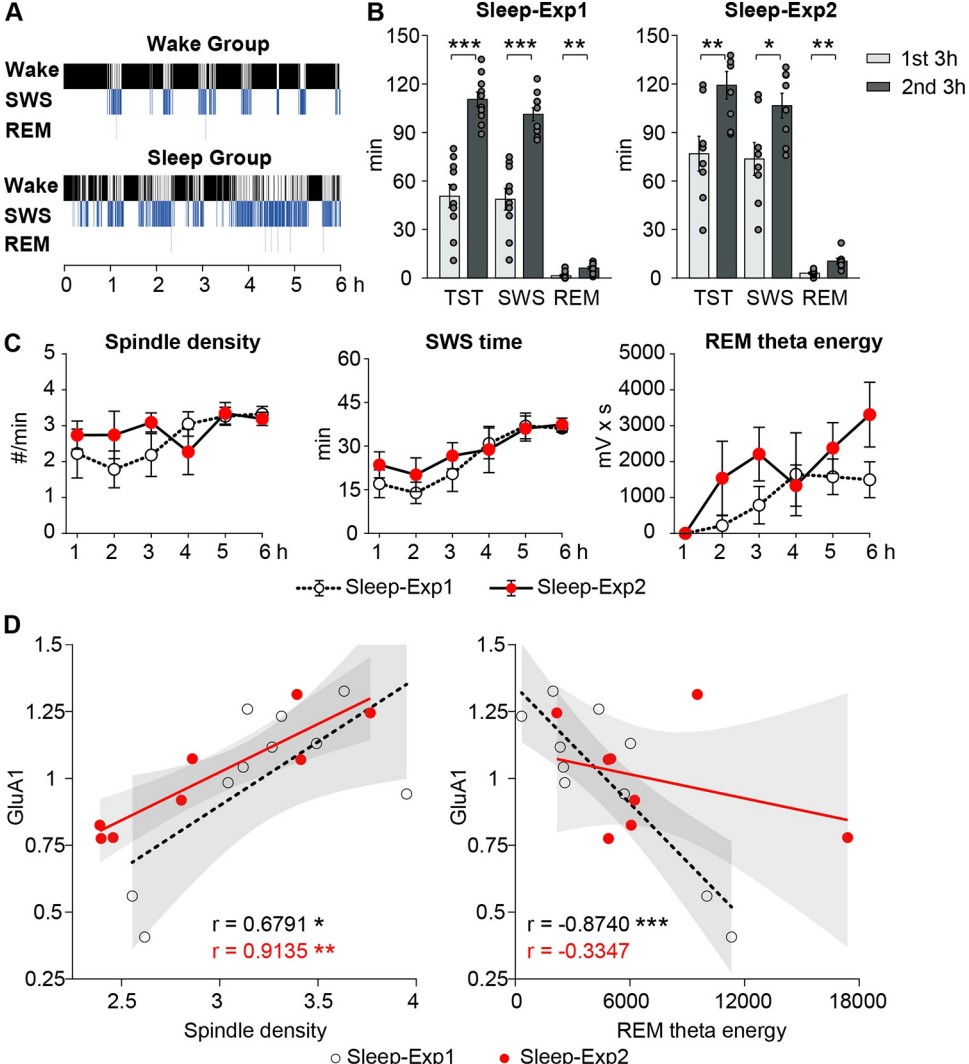

**Fig 3. Spindle density predicts GluA1 AMPAR subunit levels in the hypothalamus.** (**A**) Hypnogram for the 6-hour period before assessment of AMPAR subunits obtained in an individual rat of the Wake (top) and Sleep (bottom) group of experiment 1 (SWS, slow-wave sleep; REM, rapid eye movement sleep). (**B**) Mean (± SEM) total sleep time (TST), time spent in SWS, and time spent in REM sleep in the first and second 3 hours of the 6-hour interval before subunit assessment in the Sleep groups of experiment 1 (Sleep-Exp1, left) and experiment 2 (Sleep-Exp2, right), dot plots overlaid; *** $p < 0.001$, ** $p < 0.01$; * $p < 0.05$, for $t$ tests between first and second 3-hour intervals. (**C**) Mean (± SEM) time courses for the major sleep oscillatory signals (spindle density, time in SWS, and theta energy in REM sleep) during the 6-hour interval before subunit assessment in the Sleep groups of experiments 1 (white dots) and 2 (red dots). Sleep and oscillatory hallmarks of SWS and REM sleep were more pronounced during the second than first 3-hour interval. (**D**) Pearson product-moment correlations between levels of GluA1 subunit-containing AMPARs in hypothalamus and spindle density (left) and REM sleep theta energy (right) during the 3 hours before subunit assessment in the Sleep groups of experiment 1 (black dots and lines) and 2 (red dots and lines, grey shades, 95% confidence intervals). Data of experiment 2 were used to validate, via stepwise regression models, significant correlations identified in experiment 1. Only the positive correlation of spindle density with GluA1 subunit levels proved to be robust; the underlying data sets are available in an online supporting file (S1 Data).

for 6 hours prior to AMPAR subunit assessment. For this purpose, we subjected these parameters to stepwise linear regression analyses. Again, we found increases in total sleep time as well as SWS and REM sleep duration in the second compared to the first 3 hours of the experimental 6-hour interval (**Figs 3B, 3C,** and **S4**). Regression analyses confirmed spindle density as a

factor predicting GluA1 subunit levels in hypothalamic synaptoneurosomes (b = 19.624, SEM = 7.12, t(5) = 2.75, $p < 0.05$, for the optimal model fit): spindle density during the second 3-hour interval and GluA1 subunit levels were positively associated ($r = 0.9135$, $p = 0.0015$; **Fig 3D**). This correlation was not observed for cortical levels of GluA1 subunits ($r = -0.0563$, $p = 0.8947$, $z = 2.6669$, $p = 0.0077$ for the difference between correlations in hypothalamus and neocortex). In contrast to spindle density, the negative associations of SWS duration and REM sleep theta energy with GluA1-AMPAR subunit levels observed in experiment 1, however, were not confirmed (**Fig 3D**). Note also that not only is the statistical link between high-amplitude REM theta activity and the down-regulation of hypothalamic GluA1-containing AMPARs restricted to experiment 1, but that the absence of effects on GluA1-AMPAR regulation of selective REM sleep deprivation (compared to regular sleep) in experiment 2 indicates that SWS alone is sufficient for sleep-dependent synaptic renormalization.

## Discussion

The SHY assumes that periods of wakefulness, due to increased information encoding and processing, go along with a net increase in synaptic networks, whereas subsequent sleep supports the renormalization of net synaptic strength [1,2]. The hypothesis refers mainly to glutamatergic transmission via synaptic AMPARs as a fundamental mechanism to control synaptic strength in the major forms of synaptic plasticity [27]. Supporting SHY, wake-related increases and sleep-related decreases in AMPAR levels have been shown in previous studies in neocortical as well as hippocampal networks [8–10,28,29]. The present findings extend and refine the SHY in several ways: We provide first evidence that SHY also applies to hypothalamic networks, which, themselves, are mainly involved in the homeostatic regulation of various organismic functions, most notably energy metabolism [12,30] and the regulation of sleep and wakefulness [31,32]. Moreover, comparing effects of selective REM sleep deprivation and total sleep deprivation, our findings provide novel experimental evidence that SWS, rather than REM sleep, is the main driver of synaptic renormalization during sleep, which is in agreement with assumptions derived from previous work based mainly on correlational and computational approaches (e.g., [3,33,34]). Although our REM sleep deprivation experiments identified SWS as a whole to promote renormalization of AMPAR subunits, spindle activity as one hallmark of this sleep stage consistently predicted increased, rather than decreased, GluA1-containing AMPAR levels in the hypothalamus. This pattern is in line with the notion that spindles contribute to the maintenance of synaptic potentiation and connectivity underlying the formation of memories during sleep [35,36].

Our analyses of AMPAR subunit levels in neocortical synaptoneurosomes replicate virtually all findings of a previous study by Vyazovskiy and colleagues [8], which compared the effects of sleep and wakefulness on AMPAR levels in neocortex with those in hippocampal synaptoneurosomes. Our first experiment revealed an almost 40% decrease in GluA1-containing AMPARs after the daytime sleep period in comparison with the nocturnal wake period that is closely comparable in size to the previous findings. The magnitude of the difference in GluA1 AMPAR subunit levels after sleep versus wakefulness very well matches changes observed after learning and experimentally induced synaptic long-term potentiation (LTP) in vivo (e.g., [29,37–41]). This is consistent with the view that stimulus processing during wakefulness leads to a net increase in potentiated synapses integrating GluA1 AMPARs, whereas sleep favors the removal of these receptors and, thus, leads to a net renormalization of synaptic strength. Also, the decrease in levels of GluA1 AMPARs phosphorylated at Ser845 and of receptors phosphorylated at Ser831 after sleep replicates previous results by Vyazovskiy and colleagues [8]. Both types of phosphorylation are acutely involved in mediating synaptic LTP and its maintenance.

Phosphorylation at Ser831 mediates an increase in single-channel conductance at the AMPAR. The dephosphorylation at Ser845 decreases the probability of channel openings and promotes the internalization of the receptor during long-term depression (LTD) and the downscaling of synapses [23,24]. Dephosphorylation at Ser845 during downscaling results from a loss of protein kinase A (PKA) from synapses [42]. Finally, both the present and the previous study by Vyazovskiy and colleagues [8] agree in showing a slight sleep-associated decrease in levels of GluA2-containing AMPARs, which generally was less robust than that of GluA1 AMPARs, possibly reflecting less pronounced plasticity of GluA2-containing receptors [43]. Thus, the decrease in GluA2 subunits did reach significance in the Sleep group of our experiment 1, but not in the condition of undisturbed sleep of experiment 2 or in the respective sleep condition of the study by Vyazovskiy and colleagues [8]. Experimental induction of synaptic LTP typically increases expression of GluA2 AMPARs in parallel with GluA1 AMPARs [37,39,44,45]. The presence of the GluA2 subunit, then, renders the receptor impermeable to calcium, thus restricting receptor gating mainly to sodium and potassium ions, a function assumed to protect the neuron from excitotoxicity [46]. Overall, our results from neocortical synaptoneurosomes, in accordance with the earlier findings by Vyazovskiy and colleagues [8] and in conjunction with studies employing various other structural and functional measures of network synaptic connectivity—like the size of axon-spine interfaces and spine heads determined by electron microscopy [9] and the amplitude of somatosensory and motor evoked potentials in humans [47]—corroborate the view proposed by SHY that, in the neocortex, network synaptic potentiation increases over periods of wakefulness, whereas sleep promotes a depression and renormalization of synaptic strength.

The main finding of our experiments is that, in comparison with wakefulness, sleep induces a decrease in synaptic AMPAR levels in hypothalamic networks in virtually the same way as in neocortical networks. We did not find any statistically significant difference in the response to sleep between hypothalamic and neocortical synaptoneurosomes in any of the 4 targeted AMPAR proteins. Importantly, in hypothalamic synaptoneurosomes, we found pronounced sleep-associated decreases (in experiment 1) of levels of GluA1-containing AMPARs and of GluA1 AMPARs phosphorylated at Ser845, i.e., the proteins that displayed most robust sleep-related decreases also in neocortical synaptoneurosomes. Thus, sleep appears to rather uniformly (down-)regulate connectivity in glutamatergic synaptic networks throughout cortical and subcortical structures, not only in neocortex and hippocampus, as observed before, but also in the hypothalamus. This is notable considering that the hypothalamus is thought to be involved to a much lesser extent in classical forms of learning and memory formation and underlying synaptic plastic processes than the hippocampus and neocortex [18,48]. Moreover, in contrast to hippocampus and neocortex, the homeostatic regulation of organismic functions by hypothalamic nuclei is predominantly controlled by neuropeptides, a process that has been proposed to imply altered mechanisms of AMPAR-mediated plasticity [15]. Indeed, we have previously shown that short-term high-fat feeding in rats induces opposing changes in synaptic AMPAR levels in hypothalamus and neocortex, i.e., robust reductions in the former and signs of increases in the latter [17]. Those findings highlight that the uniform down-regulation of synaptic AMPAR levels in neocortex and hypothalamus is specifically promoted by sleep. The cellular mechanism mediating such unified down-regulation of AMPARs is presently not clear but may involve pathways involving cyclin-dependent kinase 5 (CDK5) and the immediate early gene *Homer1a* [10,34,49].

Our second experiment, in which we compared the effects of selective REM sleep deprivation and total sleep deprivation, provided firm evidence that SWS is the main driver of the sleep-dependent down-regulation of synaptic AMPAR levels in neocortex and hypothalamus. Whereas the selective deprivation of REM sleep periods in comparison to undisturbed sleep

left the levels of all 4 AMPAR proteins of interest unchanged, only total sleep deprivation—comprising deprivation of REM sleep and of SWS—induced basically the same pronounced increase in GluA1-containing AMPAR subunit levels as wakefulness compared with sleep in our first experiment. The more direct proof of a causal role of SWS by selectively depriving SWS is basically impossible because the animal enters REM sleep only after a period of SWS has occurred, so that preventing the occurrence of SWS inevitably results in total sleep deprivation. Accordingly, in the absence of any effect of selective REM sleep deprivation, the effect of total sleep deprivation can be safely taken to infer a causal contribution of SWS to the renormalization of AMPAR levels. At a first glance, our finding of unchanged AMPAR levels after REM sleep deprivation conflicts with several other studies pointing to a crucial role of REM sleep in synaptic regulation [50–52]. All of these studies made use of the "flowerpot method" to induce a preferential suppression of REM sleep [53]. However, whereas in the present study, the animals were deprived of REM sleep only for a rather short 6-hour interval (in order to dissociate effects of REM sleep from those of undisturbed sleep occurring over the same short 6-hour time period), in those experiments REM sleep was prevented for much longer intervals of up to 75 hours (of rest/activity cycles). Such prolonged REM sleep deprivation is well known to induce signs of stress (reflected by increased levels of norepinephrine and corticosteroids), which themselves strongly affect AMPAR regulation [54]. By contrast, we prevented REM sleep by gentle handling of the animals, and only for a rather short period, which can be expected to minimize stress-related confounds [55,56]. Additionally, some of the above-mentioned studies (e.g., [51]) did not differentiate between neural and glial contributions to AMPAR levels, whereas the present study focused on synaptoneurosomes specifically reflecting AMPAR levels at neuronal synapses.

While we applied sleep deprivation procedures to demonstrate a driving role of SWS in the down-regulation of AMPAR levels, we used correlational analyses to identify sleep oscillatory signatures that most likely contribute to the regulation of AMPARs during sleep. In a 2-step procedure, we first ran exploratory correlation analyses of the data of the Sleep group of experiment 1, which revealed that 3 sleep parameters (i.e., time in SWS, spindle density, and REM theta energy) were consistently associated with GluA1 levels. We then applied a stepwise regression approach to these parameters as assessed in the Sleep group of experiment 2. Only spindle density survived this hypothesis-driven approach, i.e., across both experiments increased spindle density during the 3 hours before subunit assessment predicted higher levels of GluA1 subunit levels in hypothalamic synaptoneurosomes. This positive association fits well with a body of evidence indicating an enhancing effect of sleep spindles on the consolidation of newly encoded memories, a process associated with the persistence or relative enhancement of connectivity in specific synaptic ensembles, rather than global synaptic down-regulation [57,58]. In the neocortex, spindles have been shown to underlie plastic synaptic changes mediating an augmenting response evoked by pulse trains [59]. Spindles are associated with the replay of newly encoded memories [60,61], and, in contrast to spindle-inactive cells that decrease their activity during SWS, neurons active during spindles display a relative up-regulation of their activity in the course of sleep [62]. Noteworthy, in the present experiments, a robust association with spindles was observed for GluA1-containing AMPAR levels in hypothalamic, but not in neocortical synaptoneurosomes. One reason for this might be that in the neocortex, up-regulating effects of spindles on GluA1 AMPARs are primarily conveyed by locally acting spindles [63,64], whereas we assessed the link between EEG spindles recorded at only a single electrode site and global AMPAR levels in an entire neocortical hemisphere. Moreover, in the neocortex, the number of cells that are activated during a spindle is relatively low and fluctuates significantly across consecutive SWS episodes [62]. It is therefore plausible to assume that spindles in neocortical networks only affect a relatively small fraction of

AMPARs. Our findings suggest a different scenario for hypothalamic networks, in which spindles seem to exert a more widespread effect, regulating AMPAR levels throughout the hypothalamus. While the presence of spindles in hypothalamic networks awaits confirmation, the pattern found here might well be in line with a role of hypothalamic circuitries in gating long-term memory formation [18,65]. In light of computational models predicting that temporal firing patterns during slow-wave activity favor synaptic LTD and down-regulation of synaptic connectivity [3,33], it might also surprise that we did not observe any systematic negative correlation of neocortical GluA1 subunit levels and measures of slow-wave activity. However, slow-wave activity is probably not a homogenous entity but comprises different types of waves with partly opposing effects on synaptic scaling, whose distinction is difficult when based solely on EEG criteria [66,67]. Also, the size of our samples was rather small in terms of correlation analyses, rendering any conclusion tentative, particularly so with regard to null findings.

To dissociate effects of sleep from those of the circadian rhythm, we compared AMPAR subunit levels between experiment 1 (in which sleep and wakefulness occurred at opposing phases of the circadian cycle) and experiment 2 (in which the circadian phase was kept constant across the 3 groups). These analyses confirmed that the pronounced decreasing effect of sleep on GluA1-containing AMPARs emerges independently of any circadian influence. In some contrast, circadian rhythmicity significantly contributed to the decrease in the levels of phosphorylated GluA1-containig AMPARs. These findings are in line with evidence that the phosphorylation of GluA1 AMPARs is, to a certain extent, controlled by circadian clocks and may therefore occur at least in part independently of synaptic network plasticity associated with learning and memory processes during sleep and wakefulness [68,69].

The molecular measures of AMPAR subunits obtained with western blots were collected in the absence of complementing function measurements, which is an obvious limitation of our study. We choose this approach as we aimed at a direct replication of the findings in neocortical synaptoneurosomes by Vyazovskiy and colleagues [8], which represents a key study in support of the SHY that has been confirmed by many other studies using different measures of synaptic plasticity [2,34]. Western blotting provides a valid assessment of average receptor levels in large regions like hypothalamus and neocortex but does not permit the assessment of AMPAR subunit levels in specific circuits. Therefore, a tempting open question for future research is to which extent sleep-dependent synaptic down-regulation of AMPARs pertains to neuronal circuits that genuinely enable the homeostatic regulation of organismic functions, like food intake and sleep itself.

## Materials and methods

### Animals

A total of 40 male Wistar rats aged 13 weeks (Janvier, Le Genest-Saint-Isle, France) were used for the 2 experiments. The rats were kept at controlled temperature ($22 \pm 2°C$) and humidity (45% to 65%) on a 12-h/12-h light/dark cycle with lights off at 18:00 h. Water and food were available ad libitum. All animals were habituated to their home cage and handled for 7 consecutive days (10 to 30 min/day) after arrival at the central animal facility. Animals were routinely checked by laboratory staff. Failure to groom and/or loss of more than 20% body weight were set as criteria of potential sickness and lead to the exclusion of the animal.

### Ethics statement

All experimental procedures were performed in accordance with the European animal protection laws and policies and were approved by the local animal welfare institutional review board (Regierungspräsidium Tübingen, Baden-Württemberg; # MPV 1/17).

## Experimental procedures

**Design and experimental schedules.** Experiment 1 comprised 2 groups of rats, a Sleep group and a Wake group, each including 16 animals. Each animal was habituated to the experimental recording box (dark gray PVC, 30 × 30 cm, height: 40 cm) on 3 consecutive days prior to the experiment proper, for 6 hours per day. On the fourth day, the experimental 6-hour period was performed while EEG and EMG signals were continuously recorded and the rat's behavior was videotaped for offline analyses. For the Sleep group, the recording period took place at the beginning of the light phase, i.e., between 06:00 and 12:00 h, and for the Wake group at the beginning of the dark phase, i.e., between 18:00 and 24:00 h. During the experimental 6-hour period, the animals had ad libitum access to water but were not provided food. In 6 rats of each group, EEG and EMG signals were not recorded and sleep was scored based on videotaped behavior.

Experiment 2 comprised 3 experimental groups of rats exposed to (i) TSD, or (ii) REM-D, or (iii) whose sleep was not disturbed (Sleep), each including $n = 8$ animals. Animals were habituated to the experimental 6-hour recording period as described for experiment 1. For all groups, the recording period took place between 06:00 and 12:00 h; as in experiment 1, the animals were deprived of food but not of water. Sleep deprivation in the TSD and REM-D group was implemented by "gentle handling," which involves gentle tapping on the box and, if necessary, gently shaking the box. No intense stimulation was used, and video recordings ensured that no signs of startle or freezing behavior occurred. This procedure minimizes stress and confounding influences of locomotion when applied over a longer period [70,71]. It was applied in the TSD group whenever behavior and EEG recordings indicated signs of sleep, and in the REM-D group upon the occurrence of signs of REM sleep (occurrence of EEG theta and strong reduction in muscle tone).

In both experiments, animals were deeply anesthetized with isoflurane (within 1 min) and killed by cervical dislocation immediately after the 6-hour experimental recording period. The head was cooled in liquid nitrogen and the whole brain was removed. The left cortical hemisphere and the hypothalamus were dissected, and samples were immediately frozen in liquid nitrogen and stored at −80°C for later assessment of AMPAR subunit levels.

**Surgery.** Animals were anesthetized with an intraperitoneal injection of fentanyl (0.005 mg/kg of body weight), midazolam (2.0 mg/kg), and medetomidin (0.15 mg/kg). They were placed into a stereotaxic frame and were supplemented with isoflurane (0.5%) if necessary. The scalp was exposed and 5 holes were drilled into the skull. Four EEG screw electrodes (PlasticsOne, United States of America) were implanted (2 frontal electrodes: anterior +2.6 mm, lateral ±1.5 mm; parietal electrode: posterior −2.0 mm, lateral 2.5 mm from Bregma; occipital reference electrode: posterior −10.0 mm, lateral 0.0 mm from lambda). For EMG recordings, 2 stainless steel wires (PlasticsOne) were implanted into the neck muscle. Electrodes were connected to a 6-channel electrode pedestal (PlasticsOne) and fixed with cold polymerizing dental resin, and the wound was sutured. After surgery, the animals were single-housed in their home cages and sleep recording was conducted after at least 7 days of recovery.

**Sleep recordings and classification of sleep stages.** During the 6-hour experimental recording period, the animal's behavior was continuously monitored using a video camera mounted on the recording box. The animals were connected to a commutator that compensated their movements and enabled the connection of the electrodes with the amplifier (Model 15A54, Grass Technologies, USA). EEG and EMG signals were filtered between 0.1 and 300 Hz and 30 and 300 Hz, respectively. Signals were digitalized at a sampling rate of 1,000 Hz (Power1401, Cambridge Electronic Design, United Kingdom). Recordings were visually

inspected for artifacts in Spike2 (Version 8; Cambridge Electronic Design, UK), and parameters of interest were determined as follows.

Sleep stages were scored based on EEG and EMG signals for succeeding 10-s epochs. Besides Wake, 3 sleep stages, i.e., SWS, REM sleep, and Pre-REM sleep, were determined offline using standard visual scoring procedures as previously described [72–74]. Wakefulness was identified by mixed-frequency EEG and sustained EMG activity, SWS by the presence of high amplitude low activity (delta activity: <4.0 Hz) and reduced EMG tone, and REM sleep by low-amplitude EEG activity with predominant theta activity (5.0 to 10.0 Hz), phasic muscle twitches, and minimum EMG tone. Pre-REM was identified by decreased delta activity, progressive increase of theta activity, and presence of sleep spindles. Recordings were scored by 2 experienced experimenters (interrater agreement >89.9%). Consensus was achieved afterwards for epochs with discrepant classification.

In 6 rats each of the Sleep and Wake groups of experiment 1, sleep versus wakefulness was scored solely based on behavioral criteria following standard procedures [75–77]. Sleep was scored whenever the rat showed a typical sleep posture and stayed immobile for at least 10 s. This visual scoring approach has been shown in previous rodent studies by our and other groups to consistently match conventional EEG/EMG-based scoring by more than 92%.

**EEG analyses.** EEG signals were analyzed to determine the power within the frequency bands hallmarking SWS and REM sleep, i.e., slow-wave activity (0.1 to 4 Hz) and theta activity (5 to 10 Hz), respectively. The EEG signal was filtered in the relevant frequency bands using a third-order Butterworth filter. The power measure was determined by computing the absolute value of the Hilbert-transformed filtered signal. In addition, energy within the slow-wave activity and theta frequency bands was obtained by integrating the Hilbert-transformed filtered signal over the duration of the respective SWS and REM sleep epochs.

To identify sleep spindles and SO events during SWS and Pre-REM, offline algorithms were used as described in detail previously [78–81]. For detection of spindle events, the EEG signal was filtered between 10.0 and 16.0 Hz. Then, the envelope was extracted via the absolute value, i.e., the instantaneous amplitude, of the Hilbert transform on the filtered signal. Next, we determined 3 thresholds for spindle detection based on the mean and standard deviation (SD) of the spindle band envelope during NREM sleep: the absolute value of the transformed signal exceeds 1.5 SDs (lower threshold) for at least 0.5 s but no more than 2.5 s, 2.0 SDs (middle threshold) for at least 0.25 s, and 2.5 SDs (upper threshold) at least once, respectively. Spindle onset was defined by the time when the signal exceeds the lower threshold for the first time. Spindle power was calculated as the integral of the envelope of the Hilbert-transformed signal between spindle onset and end. For calculating Hilbert transformations, the built-in function "hilbert" was used in Matlab. The envelope was extracted using the Matlab function "abs," which returns the absolute value (modulus), i.e., the "instantaneous amplitude" of the transformed signal. For each rat, the total number of spindles, spindle density (per min SWS), and the average spindle duration were determined.

For the detection of individual SO events, the EEG signal was filtered between 0.1 and 4.0 Hz, and an event was selected in the EEG if the following criteria were fulfilled: (i) 2 consecutive negative-to-positive 0 crossings of the signal occur at an interval between 0.5 and 2.0 s and (ii) the highest and lowest value are detected between every 2 of these time points (i.e., 1 negative and 1 positive peak between 2 succeeding positive-to-negative 0 crossings). Intervals of positive-to-negative 0 crossings were marked as SOs if the corresponding difference between the negative amplitude and negative-to-positive amplitude was greater than two-thirds of the average of the respective amplitude values across the whole recording. For each animal, the total number of SOs, SO density (per min SWS and PRE-REM sleep), and average SO amplitude, were determined.

### Assessment of AMPAR subunit levels

**Preparation of synaptoneurosomes.** Preparation followed previously published procedures [17]. Cortical and hypothalamic tissue was rapidly dissected and immediately homogenized in a glass Teflon homogenizer in synaptic protein extraction reagent (Syn-PER; Thermo Scientific, USA) supplemented with a protease and phosphatase inhibitor cocktail (Thermo Scientific). The homogenate was centrifuged at $1,200 \times g$ for 10 min at 4˚C to remove cell debris, and the supernatant was centrifuged at $15,000 \times g$ for 20 min at 4˚C. Subsequently, the supernatant (cytosolic fraction) was removed, and the pellets containing the synaptoneurosomes were resuspended in Syn-PER. The protein concentration of the cytosolic and synaptoneurosome fractions was determined by bicinchoninic acid assay (Thermo Scientific). After each extraction procedure, samples of the homogenate, supernatant, and synaptoneurosome samples were probed for expression of the postsynaptic marker PSD-95, to confirm enrichment of PSD-95 in the synaptoneurosome fraction (**S1 Fig**) before further processing.

**Western blotting.** Samples were heat-denatured and equal amounts (30 μg in experiment 1; 15 μg in experiment 2) of the protein sample from each animal were separated with SDS-polyacrylamide gel electrophoresis (5% (w/v) stacking and 8% separating gels) before electrophoretic transfer onto a 0.45-μm-pore nitrocellulose membrane (Carl Roth, Germany) using a semi-dry transfer system (Bio-Rad, Germany) at 0.8 mA/cm$^2$. Membranes were first blocked for 1 hour at room temperature in freshly prepared 5% powdered nonfat milk (Carl Roth) in phosphate-buffered saline (PBS) and subsequently incubated overnight with primary antibodies with agitation at 4˚C. Primary antibodies were diluted in blocking buffer containing 0.1% Tween 20 (Carl Roth) as follows: rabbit-anti-GluA1 (1:3,000), rabbit-anti-GluA2 (1:1,000), rabbit-anti-phospho-Ser845 (1:3,000), rabbit-anti-phospho-Ser831 (1:750; all Merck Millipore, Germany), mouse-anti-β-actin (1:10,000; Abcam, UK), rabbit-anti-β-tubulin (1:50,000; BioLegend, USA), mouse-anti-PSD95 (1:1,000; BD Biosciences, Germany). After several washes in PBS, membranes were incubated in HRP-conjugated anti-rabbit (1:5,000; Merck Millipore) or anti-mouse antibodies (1:4,000; BioLegend) for 2 hours. HRP activity was detected using the chemiluminescence reagents provided with the ECL kit (Thermo Scientific). Fluorescence images of the blots were obtained with a FUSION-FX7 imaging system (Vilber Lourmat, France) in experiment 1 and an Azure 600 imaging system (Azure Biosystems, USA) in experiment 2. For antibody stripping, blots were incubated in stripping solution (2% SDS, 0.8% ß-mercaptoethanol in 0.0625 M Tris-HCl (pH 6.8)) at 50˚C for 30 to 45 min with some agitation, rinsed with ultrapure water for 1 to 2 min, and subsequently washed 3 times for 5 min with PBS with 0.1% Tween.

**Image analysis.** Integrated background-subtracted (rolling-ball algorithm) signal intensity for each antibody band was quantified with ImageJ software. GluA1, GluA2, phospho-Ser845, and phospho-Ser831 bands were normalized with reference to the corresponding β-actin band in the same sample, the latter serving as loading control. To compare experimental groups, actin-normalized intensity values were normalized (in %) to the average of the values in the reference group within the same blot. To assess GluA1 phosphorylation, we first probed blots with anti-phospho-Ser845 or anti-phospho-Ser831 antibody, stripped them, and subsequently reprobed them with anti-GluA1 antibody, which recognizes both phosphorylated and nonphosphorylated GluA1. Individual AMPAR subunit levels were expressed as percent values, with the respective average levels in the Wake group (experiment 1) and Sleep group (experiment 2) set to 100%.

### Statistical analyses

Statistical analyses were performed with Matlab (R2021a; MathWorks, USA) and SPSS statistical software (IBM SPSS Statistics 24, USA). They generally relied on Student *t* tests (unpaired,

two-sided) and analyses of variance (ANOVA) with a Group factor for the different experimental groups and an Experiment factor for comparisons between experiments 1 and 2, as appropriate. Linear correlation analyses between individual AMPAR subunit levels (expressed as percent values) and sleep parameters of interest relied on Pearson product-moment correlation coefficients. Stepwise regression analysis of data from experiment 2 was employed to confirm or rebut significant correlations obtained in the analyses of experiment 1.

Differences in correlations coefficients were tested using Cocor analysis (http://comparingcorrelations.org/; [82]). All analyses were run after normal distribution of the respective data had been confirmed using Shapiro–Wilk test. In one case where normality was violated, the result of the *t* test was confirmed by an additional nonparametric test (Mann–Whitney U test). A *p*-value of $< 0.05$ was considered statistically significant.

## Supporting information

**S1 Fig. Expression of the postsynaptic marker PSD-95, of tubulin, and of ß-actin in different protein fractions.** Representative western blots showing homogenate (Ho), synaptoneurosomes (Sn), and supernatant (Su) fractions from (**A**) hypothalamus and (**D**) neocortex. (**B, E**) Quantification of PSD-95, tubulin, and ß-actin. Integrated density values were normalized to values of homogenates (set to 100%) for samples from each animal. (**C, F**) Quantification of PSD-95 and tubulin relative to ß-actin density were normalized to values of homogenates (set to 100%). Circles represent samples from individual animals (hypothalamus, *n* = 8 rats; neocortex, *n* = 15 rats); the underlying data sets are available in an online supporting file (S1 Data).
(TIF)

**S2 Fig. AMPAR levels in supernatants of hypothalamic and neocortical samples in the Sleep and Wake groups of experiment 1.** (**A**) Levels of GluA1- (left) and GluA2-containing AMPARs (right) and (**B**) of GluA1 phosphorylated at Ser845 (left) and at Ser831 (right) in hypothalamus and (**C, D**) neocortex. Mean ± SEM normalized AMPAR levels are shown with the mean for the Wake group set to 100%. On top, 2 example immunoblots are shown for each group (s1, s2, w1, w2; GluA1, GluA2, phospho-Ser845, and phospho-Ser831 bands were normalized with reference to the corresponding β-actin band in the same sample, the latter serving as loading control). There were no significant differences between groups for any measure; the underlying data sets are available in an online supporting file (S1 Data).
(TIF)

**S3 Fig. AMPAR levels in supernatants of hypothalamic and neocortical samples after undisturbed sleep (S), total sleep deprivation (TSD), and REM sleep deprivation (REM-D) in experiment 2.** (**A**) Levels of GluA1- (left) and GluA2-containing AMPARs (right) and (**B**) of GluA1 phosphorylated at Ser845 (left) and at Ser831 (right) in hypothalamus and (**C, D**) neocortex. Mean ± SEM normalized AMPAR levels are shown with the mean for the Sleep control group set to 100%. On top, 2 example immunoblots are shown for each group (s1, s2, t1, t2, r1, r2; GluA1, GluA2, phospho-Ser845, and phospho-Ser831 bands were normalized with reference to the corresponding β-actin band in the same sample, the latter serving as loading control). There were no significant differences between groups for any measure; the underlying data sets are available in an online supporting file (S1 Data).
(TIF)

**S4 Fig. Sleep architecture across the 6-hour recording intervals.** Amount of (**A**) SWS and (**B**) REM sleep in minutes during each hour of the 6-hour recording session in the Sleep and Wake groups of experiment 1 (left panels) and in the Sleep and, as applicable, REM sleep-

deprivation (REM-D) groups of experiment 2 (midline panels); SWS and REM sleep duration in minutes during the final 3 hours in the Sleep (Exp1:S and Exp2:S, respectively) and, as applicable, REM-D groups in experiments 1 and 2 (right panels); *** $p < 0.001$, unpaired $t$ tests. Note that overall, the sleep ratios are very much comparable between the respective groups of experiments 1 and 2; the underlying data sets are available in an online supporting file (S1 Data).
(TIF)

**S1 Table. Correlations between sleep parameters of interest and levels of GluA1-containing AMPARs in hypothalamus and neocortex in experiment 1.**
(DOCX)

**S1 Data. Raw data underlying, in the order of Excel sheets and, respectively, data sets in the file, Figs 1B, 1C, 1D, 1E, 1F, 2B, 3B, 3C, 3D, S1B, S1C, S1E, S1F, S2A, S2B, S2C, S2D, S3A, S3B, S3C, S3D, S4A and S4B.** In each sheet, the data are referenced to the respective sections of the figure panels (e.g., left, right).
(XLSX)

**S1 Raw Images. Original images of the western blot results.**
(PDF)

## Acknowledgments

We thank Ilona Sauter for technical assistance.

## Author Contributions

**Conceptualization:** Jan Born, Manfred Hallschmid.

**Data curation:** Jianfeng Liu, Niels Niethard.

**Formal analysis:** Jianfeng Liu, Niels Niethard.

**Funding acquisition:** Niels Niethard, Ingrid Ehrlich, Jan Born.

**Investigation:** Jianfeng Liu, Niels Niethard, Yu Lun, Stoyan Dimitrov, Ingrid Ehrlich, Jan Born, Manfred Hallschmid.

**Methodology:** Jianfeng Liu, Niels Niethard, Yu Lun, Stoyan Dimitrov, Ingrid Ehrlich.

**Project administration:** Jan Born, Manfred Hallschmid.

**Resources:** Jan Born, Manfred Hallschmid.

**Supervision:** Jan Born, Manfred Hallschmid.

**Validation:** Jan Born, Manfred Hallschmid.

**Visualization:** Jianfeng Liu, Niels Niethard.

**Writing – original draft:** Jianfeng Liu, Jan Born.

**Writing – review & editing:** Jianfeng Liu, Niels Niethard, Yu Lun, Stoyan Dimitrov, Ingrid Ehrlich, Jan Born, Manfred Hallschmid.

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
