## [Editor Report · Decision Letter 0]

5 Mar 2024

Dear Dr Born, 

Thank you for submitting your manuscript entitled "Slow wave sleep drives sleep-dependent renormalization of synaptic AMPA receptor levels in the hypothalamus" for consideration as a Research Article by PLOS Biology.

Your manuscript has now been evaluated by the PLOS Biology editorial staff as well as by an academic editor with relevant expertise and I am writing to let you know that we would like to send your submission out for external peer review.

Once your full submission is complete, your paper will undergo a series of checks in preparation for peer review. After your manuscript has passed the checks it will be sent out for review. To provide the metadata for your submission, please Login to Editorial Manager (https://www.editorialmanager.com/pbiology) within two working days, i.e. by Mar 07 2024 11:59PM.

Kind regards,

Christian

Christian Schnell, PhD

Senior Editor

PLOS Biology

cschnell@plos.org

---

## [Decision Letter · Decision Letter 1]

16 Apr 2024

Dear Dr Born,

Thank you for your patience while your manuscript "Slow wave sleep drives sleep-dependent renormalization of synaptic AMPA receptor levels in the hypothalamus" was peer-reviewed at PLOS Biology. It has now been evaluated by the PLOS Biology editors, an Academic Editor with relevant expertise, and by several independent reviewers. 

In light of the reviews, which you will find at the end of this email, we would like to invite you to revise the work to thoroughly address the reviewers' reports.

As you will see below, Reviewer 1 and 3 think that the study is very well executed and provides important insights, while Reviewer 2 has concerns about the conceptual advance and some technical aspects.

We have discussed these reports with the other reviewers and our the Academic Editor. Based on these discussions, we would encourage you to focus on fully addressing the concerns but we do not think new experimental data necessary to address Reviewer 2's concerns. 

Given the extent of revision needed, we cannot make a decision about publication until we have seen the revised manuscript and your response to the reviewers' comments. Your revised manuscript is likely to be sent for further evaluation by all or a subset of the reviewers.

**IMPORTANT - SUBMITTING YOUR REVISION**

*Re-submission Checklist*

*Published Peer Review*

*PLOS Data Policy*

*Blot and Gel Data Policy*

Sincerely,

Christian

Christian Schnell, PhD

Senior Editor

PLOS Biology

cschnell@plos.org

REVIEWS:

Reviewer #1: The work explores whether the synaptic strength reductions known to happen sleep may occur in not just regions affected by sleep, but also the hypothalamus which controls sleep. They assess protein levels and phosphorylation of synaptosomes and do indeed find that especially the GluA1 subunit and it's phosphorylation are reduced, suggesting possibly reduced synaptic transmission in that structure

The data in the first figure suffers from an obvious circadian confound but the experiment in figure 2 does not. Indeed it is not clear what figure 1 added to the story, though some use for a subset of readers may come from it. 

It is surprising that REM theta predicts GluA1 levels despite Figure 2 showing lack of import for REM. I don't think this was discussed and it should be.

Overall the work is appears to be well done and well presented. 

Reviewer #2: This manuscript covers an interesting and timely topic - namely, the neurobiological underpinnings of sleep loss-driven changes that might underlie sleep homeostasis, e.g. changes to the hypothalamus, which regulates sleep and other homeostatic processes. Critically, the experiments of this study do not constitute the first assessment of biochemical changes due to sleep loss in the hypothalamus - transcriptomic studies of this have been carried out since the mid-2000s (and using more precise methodology, see Mackiewicz et al 2007). More importantly, the present studies are using very old methods to characterize such changes - using bulk biochemistry of whole hypothalamus. More recent studies have provided better resolution about the complex effects of sleep loss on this VERY complex brain region - including single-cell transcriptomics and spatial transcriptomics. As such, there isn't any new epiphany in the present paper in terms of understanding neurobiological changes related to sleep or sleep loss. Beyond that, there are significant technical concerns that muddy the waters in terms of data interpretation, described below:

1. Experiment 1 uses circadian timepoints, so it is really impossible to make any conclusion about the relative importance of sleep vs. circadian timepoints. This is EXTREMELY significant as a caveat for studying the hypothalamus, in particular, due to the many hypothalamic structures engaged by the SCN circadian clock... which could easily drive any of the changes observed.

2. There is a large amount of variability of sleep amounts in both circadian timepoint groups in Experiment 1 (unsurprisingly). Critically, there is no attempt made to take advantage of this variability, for clarifying the data, as there could be a correlation between sleep time and the biochemical measurements.. this would actually make for a much better argument than the group differences shown.

3. In the schematic illustrations showing the region sampled for hypothalamic measures, it is striking that only a small ventral most portion of the hypothalamus seems to be highlighted. Rather, it looks like the SCN is being sampled! Is this an error in making the schematic? 

4. In the experiment where sleep is actually deprived (Experiment 2) many of the biochemical measures that were significantly different between groups (including, critically, the phosphorylations that are so important for synaptic potentiation) are NOT affected. This is true for both hippocampus AND neocortex! So.. those changes emphasized in Experiment 1 are effects of the circadian timepoints, rather than effects of sleep? If that is NOT the interpretation, what is it? It isn't clear what else could possibly make sense. So, the whole title and premise of the first part of the paper seem very misleading.

5. Figure 3D - Two very major questions here. First, why is this the first attempt at a correlation? Second, why are all the animals' data not included in this correlation (it appears to be about half of the data points from animals with PSG analysis!!)?

6. A general question - since sleep deprivation is known to alter the actin cytoskeleton, is using B-actin as a control for Westerns a very good idea? It seems like there might be a better choice, and that making the wrong choice could give a false impression.

Reviewer #3: 

This is an interesting and impactful study showing that sleep leads to overall synaptic weakening also in the hypothalamus, which is an important novel finding. Liu and colleagues also provide novel evidence that NREM sleep alone is sufficient for synaptic renormalization to occur, both in cortex and hypothalamus. The results are solid and the paper well written, pending a few clarifications and qualifications for some statements that I believe are currently too strong, as indicated below.

Major points:

Line 81: the text and legend of figure 1a mentioned "filled with sleep or wake"; in reality, in both groups the 6 hours were not "filled" with just one behavioral state; please rephrase. In fact, figure 1B shows that several rats only slept 2.5-3 hours during the first half of the day, which is quite low. Based on figure 3B, most sleep happens in the last 3 hours, which raises the question whether these animals were well adapted to the light/dark cycle. Any explanation? Since it seems that there were no selection criteria before collecting the brains (i.e. even rats that slept less than 50% of the 6 hours were used), it would be useful to report (in a table?) the % of sleep and waking in the last 3 vs 6 hours before sacrifice, to see how they differ between the day and night group. Also (see below) the authors should report % of NREM and REM sleep separately. 

The authors state that the results of " experiment 2 clearly rules out any causal contribution of REM sleep" and "SWS is the main driver" for sleep-dependent synaptic downselection. I find these results convincing. But, I am not prepared to dismiss REM sleep, especially after seeing figure 3A; if this is representative, there is little REM sleep in these rats, perhaps not surprisingly since several of these animals slept mainly in the last 3 hours (?), and the experiment is in the first half of the day; in short, I think from the current results we can conclude that NREM sleep alone is sufficient for sleep-dependent down-selection, but not that REM sleep has no role in it. 

The authors interpret the positive correlation between spindle activity and AMPAR expression in the hypothalamus as supporting evidence that spindles promote memory consolidation, but it is difficult to understand why this correlation is not found in the cortex. Spindles are thalamocortical events: why would they contribute to synaptic potentiation in the hypothalamus but not in the cortex? Indeed most spindles, like slow waves, are local, but across several hours, spindles should presumably affect most of the cortex…. How (via which anatomical pathways) would spindles reach the hypothalamus?

Other points:

Abstract, line 3: It is not correct to say that SHY "focuses on AMPA signaling"; SHY has been tested using many markers of synaptic strength, only one of which is the expression of AMPA receptors expression. Other markers include the size of the postsynaptic area measured with serial electron microscopy in axospinous (excitatory) synapses, and excitatory minis. In flies, SHY was tested using presynaptic (mainly cholinergic) markers, because the majority of synapses in flies are cholinergic. 

Abstract, line 2 and 12 (and in many places in the main text, e.g. line 26 of the introduction): "global" can be misleading because it may be interpreted as saying that according to SHY, all synapses are weakened by sleep. According to SHY, the sleep-dependent weakening of synapses is not global in the sense that it does not affect every synapse; in fact, the process is selective (hence the term sleep-dependent down-selection), but broad (general), i.e. affecting most synapses and resulting in a net decrease in synaptic strength relative to waking; the fact that the process does not affect every single synapse was shown with serial EM (de Vivo 2017) and with single synapse resolution in Miyamoto et al., 2021. 

Line 28-29: The sentence should be rephrased. SHY assumes that the net effect of waking is the strengthening of many excitatory synapses. Hundreds of studies by others have shown that this strengthening is reflected in increased expression of AMPARs. So "SHY assumes that wake encoding of information manifests itself mainly in the potentiation of excitatory synapses, which is known to result in increased numbers of ….."

Line 62, 67, 74: "scaling" should be avoided, unless there is direct evidence for actual scaling.

Line 82-84: the authors should refer here to suppl figure 3, where they provide evidence that their synaptoneurosome preparation is enriched in synaptic proteins.

Line 96: please rephrase: the cortical changes do not really "correspond" to those in the hypothalamus (GluA2 and Ser831 change in cortex only). 

Figure 2B: same issue as in figure 1B. Several S rats are sleeping around or less than 50% of the first 6 hours of the day, so it would be useful to know the distribution and % of sleep and waking in the last 3 hours before sacrifice, to see how they differ across groups.

Figure 2C: the lack of hypothalamic changes in Ser845 is a bit surprising and not discussed. 

Line 110: more correctly, the authors show that AMPAR expression is higher after total SD than after sleep, not that SD increased it 

Line 333: typo in AMPAR levels

---

## [Decision Letter · Decision Letter 2]

11 Jul 2024

Dear Jan,

Thank you for your patience while we considered your revised manuscript "Slow wave sleep drives sleep-dependent renormalization of synaptic AMPA receptor levels in the hypothalamus" for publication as a Research Article at PLOS Biology. This revised version of your manuscript has been evaluated by the PLOS Biology editors, the Academic Editor and two of the original reviewers.

Based on the reviews and on our Academic Editor's assessment of your revision, we are likely to accept this manuscript for publication, provided you satisfactorily address the following data and other policy-related requests.

* We would like to suggest a small correction to the title: "Slow-wave sleep drives sleep-dependent renormalization of synaptic AMPA receptor levels in the hypothalamus"

* DATA POLICY:

Regardless of the method selected, please ensure that you provide the individual numerical values that underlie the summary data displayed in the following figure panels as they are essential for readers to assess your analysis and to reproduce it: 1BCDEF, 2BCDEF, 3B, S1BCEF, S2ABCD, S3ABCD and S4AB.

* CODE POLICY

We require the original, uncropped and minimally adjusted images supporting all blot and gel results reported in an article's figures or Supporting Information files. We will require these files before a manuscript can be accepted so please prepare and upload them now. Please carefully read our guidelines for how to prepare and upload this data: https://journals.plos.org/plosbiology/s/figures#loc-blot-and-gel-reporting-requirements

We expect to receive your revised manuscript within two weeks. 

*Published Peer Review History*

*Press*

Sincerely,

Christian

Christian Schnell, PhD

Senior Editor

cschnell@plos.org

PLOS Biology

Reviewer remarks:

Reviewer #1: [No further comments]

Reviewer #3: I thank the authors for addressing all my previous comments. I have no further suggestions.

---

## [Editor Report · Decision Letter 3]

25 Jul 2024

Dear Jan,

Thank you for the submission of your revised Research Article "Slow-wave sleep drives sleep-dependent renormalization of synaptic AMPA receptor levels in the hypothalamus" for publication in PLOS Biology. On behalf of my colleagues and the Academic Editor, Guang Yang, I am pleased to say that we can in principle accept your manuscript for publication, provided you address any remaining formatting and reporting issues. These will be detailed in an email you should receive within 2-3 business days from our colleagues in the journal operations team; no action is required from you until then. Please note that we will not be able to formally accept your manuscript and schedule it for publication until you have completed any requested changes.

When you attend to these requests, please also make sure to indicate in the corresponding figure legends when the same loading controls are used repeatedly (once for the anti-phospho antibodies and once for the anti-GluA1 antibodies). This is fine to do but it would be flagged if the paper would be screened for duplications and without reading the methods, it might not be obvious to readers why this is appropriate.

PRESS

Sincerely, 

Christian

Christian Schnell, PhD

Senior Editor

PLOS Biology

cschnell@plos.org